# Sector of Employment and Mortality: A Cohort Based on Different Administrative Archives

**DOI:** 10.3390/ijerph20105767

**Published:** 2023-05-09

**Authors:** Lisa Bauleo, Stefania Massari, Claudio Gariazzo, Paola Michelozzi, Luca Dei Bardi, Nicolas Zengarini, Sara Maio, Massimo Stafoggia, Marina Davoli, Giovanni Viegi, Alessandro Marinaccio, Giulia Cesaroni

**Affiliations:** 1Department of Epidemiology–Lazio Regional Health Service, ASL Roma 1, 00147 Rome, Italy; l.bauleo@deplazio.it (L.B.); p.michelozzi@deplazio.it (P.M.); m.davoli@deplazio.it (M.D.); 2Department of Occupational and Environmental Medicine, Epidemiology and Hygiene, Italian National Institute for Insurance against Accidents at Work (INAIL), 00143 Rome, Italy; s.massari@inail.it (S.M.); c.gariazzo@inail.it (C.G.);; 3Department of Statistical Sciences, Sapienza University of Rome, 00185 Rome, Italy; 4Regional Public Health Observatory (SEPI), ASL TO3, 10095 Grugliasco, Italy; 5Institute of Clinical Physiology, CNR, 56124 Pisa, Italy

**Keywords:** mortality, occupational sector, cohort study, occupational risk factor, routinely collected health data, administrative data

## Abstract

Administrative data can be precious in connecting information from different sectors. For the first time, we used data from the National Social Insurance Agency (INPS) to investigate the association between the occupational sectors and both non-accidental and accidental mortality. We retrieved information on occupational sectors from 1974 to 2011 for private sector workers included in the 2011 census cohort of Rome. We classified the occupational sectors into 25 categories and analyzed occupational exposure as ever/never have been employed in a sector or as the lifetime prevalent sector. We followed the subjects from the census reference day (9 October 2011) to 31 December 2019. We calculated age-standardized mortality rates for each occupational sector, separately in men and women. We used Cox regression to investigate the association between the occupational sectors and mortality, producing hazard ratios (HRs) and 95% confidence intervals (95%CI). We analyzed 910,559 30+-year-olds (53% males) followed for 7 million person-years. During the follow-up, 59,200 and 2560 died for non-accidental and accidental causes, respectively. Several occupational sectors showed high mortality risks in men in age-adjusted models: food and tobacco production with HR = 1.16 (95%CI: 1.09–8.22), metal processing (HR = 1.66, 95%CI: 1.21–11.8), footwear and wood (HR = 1.19, 95%CI: 1.11–1.28), construction (HR = 1.15, 95%CI: 1.12–1.18), hotels, camping, bars, and restaurants (HR = 1.16, 95%CI: 1.11–1.21) and cleaning (HR = 1.42, 95%CI: 1.33–1.52). In women, the sectors that showed higher mortality than the others were hotels, camping, bars, and restaurants (HR = 1.17, 95%CI: 1.10–1.25) and cleaning services (HR = 1.23, 95%CI: 1.17–1.30). Metal processing and construction sectors showed elevated accidental mortality risks in men. Social Insurance Agency data have the potential to characterize high-risk sectors and identify susceptible groups in the population.

## 1. Introduction

Worldwide, around 340 million occupational accidents and 160 million victims of work-related illnesses are reported annually. The International Labour Organization revises these estimates at intervals, and the updates indicate an increase in accidents and ill health [1]. According to a recent report about the global work-related burden of disease and injury, in 2016, 0.36 million deaths (19%) of work-related deaths were due to injuries, while the great majority, 1.52 million (81%), were due to work-related diseases [2,3]. In Italy, the Italian National Institute for Insurance against Accidents at Work (Inail) collects data about injuries and diseases caused by work activity [4]. In 2019, Inail collected 644,479 claims for compensation due to occupational injuries, 1224 of which caused death. In the same year, there were 61,198 claims for compensation due to occupational diseases in Italy. Italian employers must identify risks at work and adopt measures to prevent occupational injuries and diseases following national rules and guidelines reported in a framework law. Proper mitigation and prevention plans can manage well-known risks.

Occupational mortality statistics, available in Italy as in all European Countries, are essential for monitoring the phenomenon and allowing international comparisons. Still, they do not show the overall picture, particularly for occupational diseases. The effects of occupational exposure can even appear after retirement due to a lag effect, and it can be difficult to associate a disease or a cause of death with a specific occupation. In particular, new and emerging occupational risks, such as for tumors with low etiologic fraction, are not easily identified and consequently managed in proper mitigation and prevention procedures. Specific working sectors might experience these risks, and health outcomes due to harmful substance exposure are hardly disentangled from those of the general population by using hospitalization and mortality data. Incomplete statistics and the lack of reliable occupational data related to specific health outcomes are of primary concern when new and emerging risks have to be identified. A case-control study design can help highlight such new outcome–exposure relationships for suspected work activities. Conversely, administrative data, such as on hospitalization and mortality, linked to occupational data allow representative population-based cohort studies.

Studies in northern Europe linked population registries with census data and provided evidence of the association between the sector of employment and mortality [5]. More recently, a network of longitudinal administrative studies was established to monitor socioeconomic inequalities in health by analyzing metropolitan census cohorts in Italy [6]. Based on the 2001 national population census, the Rome Longitudinal Study allowed the investigation of mortality inequalities by occupational status and type of job in men and women [7] and, more recently, showed differences in mortality between temporary and permanent workers by sector of employment [8]. Nevertheless, unlike the 2001 census, the 2011 census of the population had not included detailed questions on the job for the whole population. Instead, it restricted the job-related questions, including those on the employment sector, to a representative sample of the resident population. Moreover, the information about occupational status included in the census refers to a picture of the census reference day. Consequently, the lack of crucial information such as past working sectors, the mansion, or the employment duration limits the possibility of investigating work-related occupational risks, especially new and emerging ones. In particular, the effect of occupational exposure to carcinogens is cumulative, and the latency of the diseases (the time elapsed between the beginning of exposure and the onset of the disease) might be very long (e.g., more than 30 years for mesothelioma). Hence, there is the need to evaluate subjects’ working history retrospectively and not only the occupational status at a point in time.

Finally, the 2011 census was the last of the entire Italian population. In fact, on October 2018, the Italian Institute of Statistics (ISTAT) started to conduct a permanent census of the Population and Housing, performing a sample survey each year to dynamically collect the main characteristics of the Italian resident population and its socioeconomic conditions at national, regional, and local levels. The permanent census of Population and Housing does not involve all 59 million Italians distributed in 24.6 million households anymore but a sample every year: about 1,400,000 households located in 2800 Italian municipalities [9]. In the absence of reliable work-related information from the census of the entire population, other sources of information are needed. The main goal of our study was to investigate the feasibility of using the information on employment sectors from the National Social Insurance Agency (INPS) for occupational epidemiology studies. In particular, we studied the association between the employment sector and mortality, considering several confounders.

## 2. Materials and Methods

### 2.1. Setting, Study Design, and Data Sources

The setting of this study was Rome, the Italian capital city located in the Latium region in Central Italy, with an average population of about 2.6 million.

We conducted a retrospective cohort study within the BIGEPI project (Uso di BIG data per la valutazione degli Effetti sanitari acuti e cronici dell’inquinamento atmosferico nella Popolazione Italiana, i.e., use of BIG data for the evaluation of the acute and chronic health effects of air pollution in the Italian population) [10].

We used the Rome Longitudinal Study, which uses a subset cohort of residents in Rome included in the Latium Region Longitudinal Study. The Latium Region Longitudinal Study is the 2011 census cohort of all Latium residents, obtained by the record linkage of 2011 census data with the Regional Health Information System, which comprises all health administrative databases, including the mortality registry [11,12]. Each subject is followed from the census reference day (9 October 2011) to 31 December 2019, the date of migration out of Rome, or death, whichever comes first.

We used data from the INPS database to enrich the Rome Longitudinal Study with information on occupational sectors from 1974 to 2011 available for private sector workers included in the cohort. INPS occupational data regard persons who worked in private companies having at least one employee from 1974 onwards [13]. The data consists of about 55% of the Italian workforce and do not include data on public employment, self-employment, artisans, domestic workers, para-subordinate workers, and occasional workers. Available information for the entire working history of a worker is the periods of employment with the economic activity of the company where the worker was employed. We classified economic activities according to the Statistical Classification of Economic Activities in the European Community, NACE Rev. 2, grouped into broader categories [14]. The linkage between the personal identifiers within the Latium Region Longitudinal Study and the INPS data was carried out at the individual level using the tax identification code by authorized personnel. The linked dataset was then anonymized, stored, and processed on Latium region servers under strict controls to protect personal data. The record linkage was performed according to the National Statistical Program, which is approved annually by the Italian Data Protection Authority [15].

### 2.2. Study Population

In the BIGEPI project, the study population was selected from the Rome Longitudinal Study to investigate the association between long-term air pollution exposure and health status. In this study, we further selected all 30–100-year-olds who had worked for at least one year in a private occupational sector. Figure 1 shows the flowchart of the selection process.

### 2.3. Occupational Data

We classified the occupational sectors into 25 categories: agriculture, forestry, and fishing; steel industry; printing and publishing; the pharmaceutical and chemical industry; manufacturing; manufacture of textiles; electricity, gas, steam, and air conditioning supply; food and tobacco production; the non-metal mineral industry; the glass-ceramic industry; metal processing; manufacture of electrical equipment; footwear and wood; construction; wholesale and retail trade; hotels, camping, bars, and restaurants; transportation and storage; insurance activities; healthcare; services; washing and dry-cleaning of textiles; waste collection, treatment, and disposal activities; hairdressing salons; cleaning services; and, finally, gas stations. First, we modeled occupational exposure as “ever/never been employed” in each sector. Second, we calculated the lifetime prevalent sector as the sector with the highest duration of employment among all the sectors in which the subject worked from 1974 to 2011.

### 2.4. Outcomes and Other Variables

We chose non-accidental and accidental mortality as the outcomes of interest. We defined non-accidental deaths as all deaths with a cause of death classified using the International Classification of Diseases, 9th Revision, Clinical Modification (ICD-9) codes 001.x-799.x and accidental deaths as all the ICD-9CM codes 800.x-999.x.

We considered several variables available from the 2011 census: sex, age, level of education (classified as high, university; medium, high school; and low, primary and junior high school), place of birth (Rome, Latium region, northern Italy, central Italy, southern and insular Italy, and other countries), marital status (single, married, separated or divorced, and widowed).

### 2.5. Statistical Analyses

We first conducted a descriptive analysis, showing the characteristics of the study population by employment sector, including the working duration (how long a person worked in a specific sector). Then, we calculated the person-time at risk for each participant from the 9th of October 2011 to the 31st of December 2019, the date of migration from Rome, or death, whichever comes first. We calculated the crude mortality rates (per 10,000) for each employment sector in men and women. We then calculated the age-standardized mortality rates (per 10,000) with 95% confidence intervals (95%CI) for each sector in men and women, using the 2011 Italian population as the standard population.

We then used Cox regression models with age as the time scale to study the association between the sector of employment (ever/never) and mortality in men and women, producing hazard ratios (HRs) and 95%CI. We performed both age-adjusted and fully adjusted models, considering the different baseline risks for educational level, marital status, and place of birth.

Finally, we replicated the descriptive and statistical analyses using the lifetime prevalent sector. In survival analysis, when we used the lifetime prevalent sector as exposure, the sectors were mutually exclusive, and we had to choose a reference category. We chose as the reference the insurance sector, a sector characterized by white-collar workers and low age-standardized mortality rates.

### 2.6. Sensitivity Analysis

Using the lifetime prevalent sector as exposure, we conducted a sensitivity analysis to investigate the role of employment duration. We restricted the analysis to those who worked for more than ten years and to those who worked for less than ten years in a sector, under the hypothesis that the strength of the association between the sector of employment and mortality increases as the length of employment increases.

## 3. Results

### 3.1. Characteristics of the Study Population

The study population comprised 478,199 men and 432,360 women, with 7 million person-years of follow-up. Men were slightly older and less educated than women, the mean age was 54 (sd 14) for men and 51 (sd 13) for women, and the percentage of low education was 41% in men and 34% in women. Compared to the population of Rome, analyzed in the BIGEPI project, our study population representing those working for at least one year in the private sector was younger and predominantly composed of men. We selected 61% of the men and 45% of the women of the original population.

Appendix A shows the characteristics of the study population by sector of employment. The most represented sectors among men were wholesale and retail trade (21%), insurance activities (18%), and construction (16%), while among women, they were services (26%), wholesale and retail trade (24%), and insurance activities (21%). In both genders, agriculture, forestry, and fishing was the sector with the oldest mean age (in men, 66 years, sd 13, and in women, 67 years, sd 12), while waste collection, treatment, and disposal activities had the youngest mean age, with 46 years (sd 12) and 44 years (sd 10) in men and women, respectively. Among men, the percentage of low education was 41% and varied from 20% in insurance to 74% in agriculture, forestry, and fishing. The percentage of people born abroad was higher than the mean (7%) in the manufacture of textiles (12%), construction (11%), and hotels, bars, and restaurants (20%). The percentage of married individuals was similar throughout the sectors. Among women, the highest percentage of low education was observed in the agriculture, forestry, and fishing sector (83%) and hairdressing salons (80%), and the lowest was observed in the steel industry (14%); overall, in all women under study, it was 34%. The percentage of women with high education was 19%. Women employed in the hotels, camping, bars, and restaurants sector were more likely born abroad (15%) than the general female population (6%).

Figure 2 reports the box whisker plots with the median and interquartile range of the duration of employment in men (above) and women (below) by sector. The minimum duration was one year by definition for each sector because we considered as exposed to a sector those who worked in the sector for at least one year. The employment duration can reflect the age of the workers in a sector.

Table 1 shows the cohort characteristics by employment sector: the number of workers, the person-years of follow-up, the number of deaths during the follow-up, and the crude mortality rates per 10,000 for non-accidental and accidental mortality in men and women. During the 7 million person-years of follow-up in men, there were 40,278 deaths from non-accidental causes and 1751 deaths from accidental causes. Among men, the crude non-accidental mortality rates varied from 60 deaths per 10,000 in workers in the waste collection, treatment, and disposal sector to 290 deaths per 10,000 in the agriculture, forestry, and fishing sector. In contrast, the crude accidental mortality rates varied between 3 per 10,000 in manufacturing or dry-cleaning of textiles and 11 per 10,000 in agriculture, forestry, and fishing. In women, during the follow-up, there were 18,922 deaths from non-accidental causes and 809 deaths from accidental causes. In women, the crude mortality rate varied from 30 deaths per 10,000 in the waste collection, treatment, and disposal sector to 190 deaths per 10,000 in the agriculture, forestry, and fishing sector.

### 3.2. Results from Statistical Analyses Using Ever/Never Worked in a Sector as Exposure

Figure 3 shows the age-standardized mortality rates with 95%CI by employment sector in men and women. The age-standardized non-accidental mortality rates per 10,000 were 814 (95%CI: 804–824) and 524 (95%CI: 515–533) in men and women, respectively. Although some sectors showed age-standardized mortality rates with wide confidence intervals due to the small number of workers in such sectors, Figure 3 highlights the sectors characterized by higher mortality compared to the mortality of the entire study population: food and tobacco production; footwear and wood; building construction; hotels, bars, and restaurants; textile washing; and cleaning services in men. In women, those with a mortality higher than the mean were the workers in hotels, bars, and restaurants, in cleaning services, and in gas stations. The age-standardized accidental mortality rates per 10,000 were 37 (95%CI: 35–39) and 24 (95%CI: 22–27) for men and women, respectively.

Table 2 shows the results of the association between the employment sector and non-accidental or accidental mortality, separately for men and women. The HR for a specific sector represents the higher (or lower) risk for those who have worked for at least one year in the sector compared to those who have not.

In men (Table 2), the age-adjusted hazard ratios for non-accidental mortality were higher than those for workers in food and tobacco production (HR = 1.16, 95%CI: 1.09–1.23), metal processing (HR = 1.66, 95%CI: 1.21–2.27), the footwear and wood sector (HR = 1.19, 95%CI: 1.11–1.18), construction (HR = 1.15, 95%CI: 1.12–1.18), wholesale and retail trade (HR = 1.06, 95%CI: 1.03–1.09), hotels, camping, bars, and restaurants (HR = 1.16, 95%CI: 1.11–1.21), and cleaning services (HR = 1.42, 95%CI: 1.33–1.52). After adjustment for marital status, education, and place of birth, the HR slightly decreased and remained statistically significant for food and tobacco production (HR = 1.11, 95%CI: 1.05–1.18), metal processing (HR = 1.48, 95%CI: 1.08–1.14), the footwear and wood sector (HR = 1.10, 95%CI: 1.02–1.18), construction (HR = 1.11, 95%CI: 1.08–1.48), hotels, camping, bars, and restaurants (HR = 1.09, 95%CI: 1.05–1.14), and cleaning services (HR = 1.25, 95%CI: 1.17–1.34). It is worth noting that in men, five sectors showed age-adjusted HRs below one, indicating a protective effect compared to all other sectors including the pharmaceutical and chemical industry; the electricity, gas, steam, and air conditioning supply sector; electrical equipment manufacturing; insurance activities; services; and hairdressing. In fully adjusted models, only three sectors had a protective association, highlighting the role of potential confounders, such as education and place of birth. The number of deaths for accidental causes was limited. However, men in metal processing and construction had higher accidental mortality risks than others. When, in addition to age, we adjusted for educational level, marital status, and place of birth, the higher risk for accidental mortality in the construction sector was no longer statistically significant.

Among women (Table 2), there was a higher risk of non-accidental mortality in workers in the sector of hotels, camping, bars, and restaurants and in the sector of cleaning services, with age-adjusted HR = 1.17 (95%CI: 1.10–1.25) and HR = 1.23 (95%CI: 1.17–1.30), respectively. In fully adjusted models, the HRs slightly decreased but still were statistically significant. There was no evidence of an association between the employment sector and accidental mortality in women.

### 3.3. Results from Statistical Analyses Using the Lifetime Prevalent Sector as the Exposure Variable

When we replicated the analyses using the lifetime prevalent sector as the exposure variable, the results, shown in the Appendix A, were very similar to those already reported.

Appendix A shows the characteristics of the study population by lifetime prevalent sector, and Appendix A reports the duration (in months) of the prevalent lifetime employment.

Appendix A show the number of deaths with crude and age-standardized (non-accidental and accidental) mortality rates per 10,000 by the lifetime prevalent employment sector in men and women, respectively. In men, the sectors with an age-standardized mortality rate for non-accidental causes larger than the age-standardized mortality rate of the entire study population (814 per 10,000, 95%CI: 804–824) were food and tobacco production; footwear and wood; construction; hotels, camping, bars, and restaurants; and finally, cleaning services. The only sector with a statistically significantly higher rate than the general population for accidental mortality was hairdressing. However, the age-standardized mortality rate of 105 per 10,000 (95%CI: 43–258) was based on only 6 deaths. In women (Appendix A), the sectors with age-standardized non-accidental mortality rates higher than the mean (525 per 10,000, 95%CI: 515–533) were hotels, camping, bars, and restaurants and cleaning services. The small number of deaths for accidental causes did not allow for highlighting particular differences in accidental mortality between sectors.

Appendix A show the results of survival analyses for men and women, respectively. Accounting for the differences in age, educational level, marital status, and place of birth, compared to workers in insurance activities, the sectors with higher risks of non-accidental mortality for men were manufacturing (HR = 1.11, 95%CI: 1.02–1.20), food and tobacco production (HR = 1.15, 95%CI: 1.07–1.24), metal processing (HR = 1.75, 95%CI: 1.14–2.69), footwear and wood (HR = 1.15, 95%CI: 1.05–1.26), construction (HR = 1.12, 95%CI: 1.08–1.17), hotels, camping, bars, and restaurants (HR = 1.12, 95%CI: 1.06–1.19), healthcare (HR = 1.15, 95%CI: 1.04–1.27), and cleaning services (HR = 1.27, 95%CI: 1.16–1.39). When we considered education, marital status, and place of birth, there was no evidence of differences in accidental mortality between sectors (Appendix A).

In women (Appendix A), taking into account the individual characteristics (age, education, place of birth, and marital status), compared to those employed in insurance activities, those working in hotels, camping, bars, and restaurants (HR = 1.11, 95%CI: 1.02–1.21), and those employed in cleaning services (HR = 1.20, 95%CI: 1.11–1.29) had a higher non-accidental mortality risk. The steel industry was the only sector with accidental mortality higher than that of the insurance sector when individual characteristics of women were taken into account.

### 3.4. Results from Sensitivity Analyses

We performed sensitivity analyses considering the lifetime prevalent sector as exposure. Specifically, we performed the survival analysis on those with at least ten years of exposure and those with less than ten years of work in the sector. The association between the employment sector and non-accidental mortality in those highly exposed (more than ten years of exposure) was sometimes more robust than in those exposed for less than ten years. For example, it was the case of men employed in the food and tobacco production sector: compared to workers in the insurance sector, the hazard ratio of those highly exposed was 1.18 (95%CI: 1.07–1.29). In contrast, the hazard ratio of the food and tobacco sector compared to the insurance sector in those exposed for less than ten years was HR = 1.05 (95%CI: 0.92–1.20). Compared to the reference sector, men employed in cleaning services for at least ten years showed an HR = 1.28 (95%CI: 1.14–1.44) versus HR = 1.19 (95%CI: 1.03–1.38) in those employed for less than ten years. The association between the employment sector and non-accidental mortality was statistically significant only for those employed for ten years or more in hotels, camping, bars, and restaurants and the health sector. In women, there was a statistically significant association between cleaning services and non-accidental mortality only in those with at least ten years of exposure.

## 4. Discussion

This study explored the possibility of using information from different administrative sources to investigate the association between occupation and health by using non-accidental and accidental mortality as the outcomes. We investigated 25 employment sectors. In men, the sectors characterized by high non-accidental mortality were food and tobacco production, footwear and wood, building construction, the hospitality industry, and washing and dry-cleaning. In women, the high-risk sectors were the hospitality industry (hotels, bars, and restaurants), washing and dry-cleaning, and gas stations. The small figures for accidental deaths did not allow us to highlight risky sectors. However, there was a suggestion for high accidental mortality risks among men in food and tobacco production, metal processing, and construction. The results were robust when considering different statistical approaches (age-standardized mortality rates and survival analysis) and exposure definitions (ever/never worked in a specific sector vs. the prevalent lifetime employment sector).

Occupational risks are related to carcinogens, asthmagens, secondhand smoke, gases, fumes, particulate matter, noise, and risk factors for injuries or low back pain [16]. A review of chemical and biological work-related occupational risks in Europe identified four major occupational groups as highly exposed to these hazards (technicians, operators, agricultural workers, and workers in elementary occupations) and several sectors associated with these exposures: production or application of pigments, resins, pesticides, adhesives, and cleaning products; agriculture; food processing and metallurgy; production of rubber, pharmaceuticals, cosmetics, plastic, and textiles [17]. Our results confirmed the high mortality risks in some industrial sectors or in construction but did not show any association in other industrial sectors typically associated with health risks. In the case of the pharmaceutical industry, for example, we did not find an association with mortality, but the pharmaceutical industry in Rome is dedicated to sales more than pharmaceutical production; in fact, it is characterized by a high level of education. Regarding specific sectors, construction workers have always had a high risk of occupational illnesses, even though their health has improved in the last 60 years thanks to medical screening programs and personal protection equipment [18]. Similarly, the sector of hotels, bars, and restaurants is characterized by environmental risk factors such as indoor air pollution [19], shift work, and exposure to cleaning products; in fact, the hospitality industry shows an excess mortality risk in both men and women.

Most publications reporting health risks related to specific jobs are based on ad hoc categories of workers [18,20,21]. However, some population-based studies used administrative databases and census data to analyze the differences in mortality, particularly cancer mortality, across employment sectors [5]. The majority of the studies that used administrative databases are from European Nordic countries, where there is a long tradition of registry-based research [22,23,24]. Our results give an idea of the potentiality of the approach. Given the large administrative databases, which may include entire municipalities or regions, it is possible to investigate specific and rare health outcomes and generate hypotheses on possible harmful exposures to be tested with analytical designs. For the same reasons, administrative data linkage allows investigation of women’s occupations, usually understudied in occupational or social epidemiology. In addition, it overcomes the lack of individual census data on the employment sector. Finally, it is a methodology that can be used in different settings and countries to monitor work-related risks over time. In the current analyses, we used two methods of exposure assessment. First, we considered a binary variable for each sector, indicating whether the person had worked for at least one year in that sector. Then, we considered the lifetime prevalent sector of employment. The two approaches have pros and cons. The first allows us to estimate the association between sector and mortality, adjusted for exposures to other sectors. The second allows the comparison of the hazard with an unexposed sector—in our case, insurance activities. However, the results were strongly comparable.

Although the United Nations SDGs have no specific target for occupational health, Target 8.8 mentions to “Protect labour rights and promote safe and secure working environments for all workers, including migrant workers, in particular women migrants, and those in precarious employment” [25]. In this context, it is paramount to have health data to conduct analytical epidemiological studies, monitor excess occupational risks, evaluate the association between occupation and health in migrants and women, and study the effects of precarious employment.

This study has some strengths and limitations. First, it confirmed known associations between occupations and deaths, indicating the usefulness of record linkage procedures between administrative archives to retrieve job information. The high sample size allowed us to investigate the association between sectors and health in men and women separately. Finally, the occupational data permitted us to trace past work history and not only cross-sectional information related to a specific point in time. A limitation of this approach is that occupational administrative data cannot adequately characterize occupational exposure profiles. INPS data report the industrial sector but do not contain information on exposure intensity and workers’ roles in a specific sector. Moreover, we considered workers with at least one year of work in a specific sector as exposed, with a possible misclassification of exposure. Regarding the Rome Longitudinal Study with the census administrative cohort of residents in Rome, it does not contain information on lifestyle risk factors, such as smoking, diet, physical activity, and exposure to indoor air pollution, which might confound the association between sectors of employment and health conditions. In addition, we could attribute the employment sector only to those who had worked in private companies. As a result, we could not analyze important sectors, such as public transportation or public healthcare, a substantial component within the Regional Health Service, a universal care system. On the other hand, we could investigate specific industrial sectors such as the steel industry, footwear and wood, food and tobacco production, construction, the hospitality industry, and washing and dry-cleaning, which are private sectors. Furthermore, the approach we used, constructing the occupational history at baseline considering all the jobs until the census reference day, does not allow us to analyze the effect of employment duration. Duration might be related to age and health status, with frail individuals less prone than others to work in harmful sectors. Finally, Rome is the Italian capital, where all regional and national institutions are located. The proportion of workers in public administration in Rome is the highest in Italy; hence, a similar approach including other provinces characterized by a higher presence of industries would be valuable.

The study investigated the association between occupational sectors and accidental and non-accidental mortality. In our view, the results have the potential to confirm work-related outcomes and to identify emerging ones, adding value to epidemiological surveillance. Significant findings will help to monitor and control the spread of diseases among workers as well as help the development of appropriate preventive plans. Further, the information derived from our study may enhance the need for the collection of thorough occupational history within the clinical management of diseases.

## 5. Conclusions

Record linkage procedures using different administrative data sources represent a precious opportunity for occupational research to generate hypotheses to be further explored with ad hoc occupational epidemiological studies.

## Figures and Tables

**Figure 1 ijerph-20-05767-f001:**
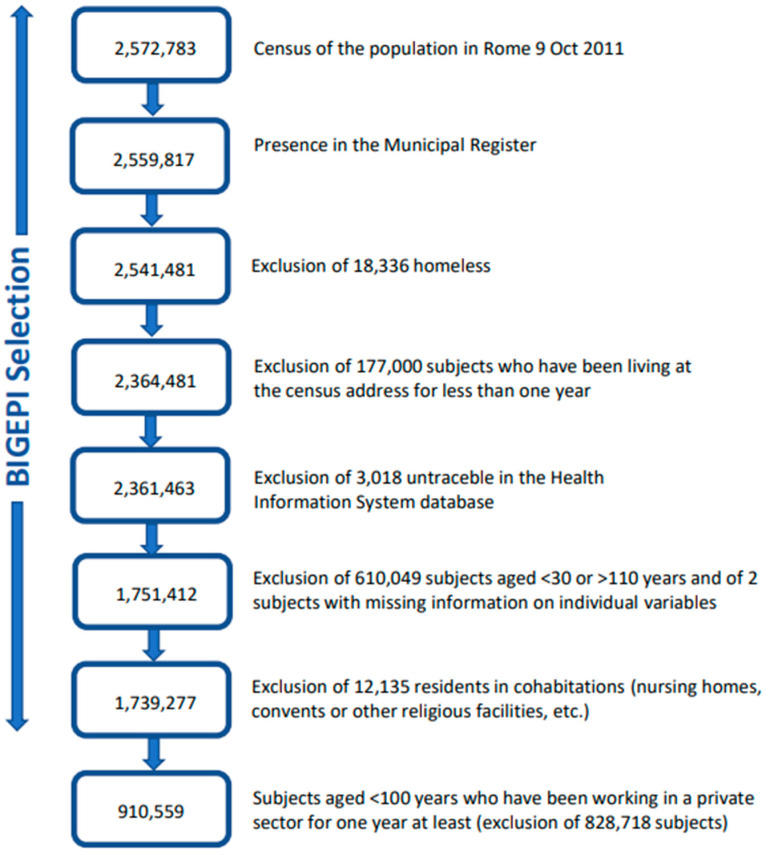
Flowchart of the study population selection.

**Figure 2 ijerph-20-05767-f002:**
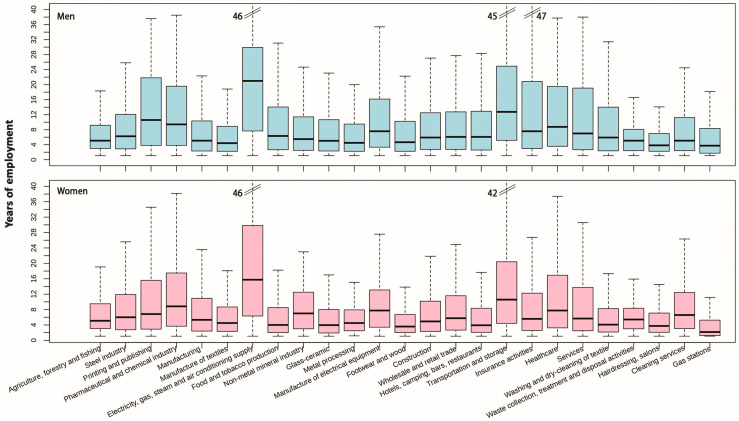
Duration of employment (in years) by sector. Rome 2011, men (**above**) and women (**below**).

**Figure 3 ijerph-20-05767-f003:**
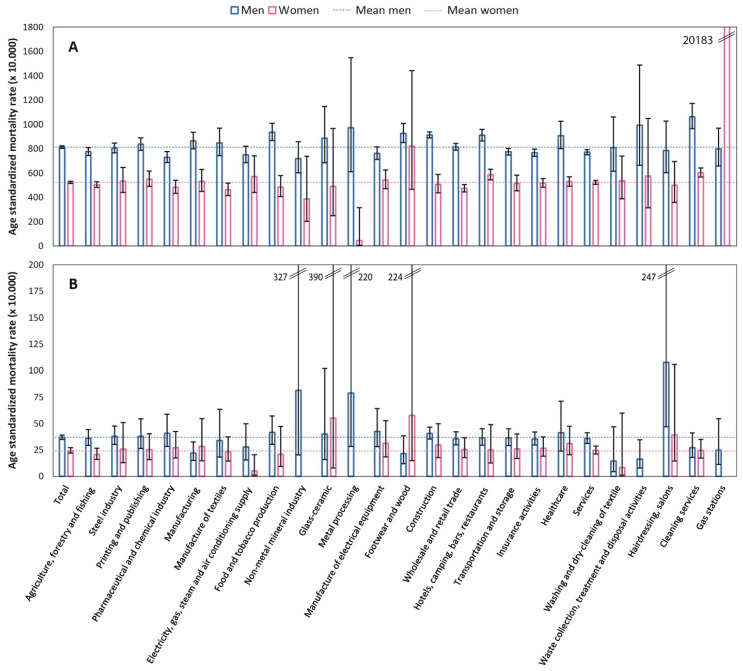
Age-standardized mortality rates (per 10,000) for non-accidental (**A**) and accidental (**B**) mortality by sector of employment. Men and women, Rome 2011–2019.

**Table 1 ijerph-20-05767-t001:** Characteristics of the cohort, number of subjects (N), person-years (PY), number of deaths, and crude mortality rates (CMR, per 10,000). Rome 2011–2019.

	Men						Women					
	N	PY	Non-Accidental		Accidental		N	PY	Non-Accidental		Accidental	
			Deaths	CMR	Deaths	CMR			Deaths	CMR	Deaths	CMR
Total	478,199	3,663,919	40,278	110	1751	5	432,360	3,399,940	18,922	56	809	2
Agriculture, forestry, and fishing	22,595	143,105	4147	290	158	11	28,599	188,769	3584	190	147	8
Industry												
Steel industry	72,872	495,713	3290	66	185	4	29,807	207,626	462	22	23	1
Printing and publishing	17,140	114,046	1611	141	56	5	11,573	79,714	478	60	21	3
Pharmaceutical and chemical industry	17,438	115,807	1472	127	64	6	11,161	76,884	454	59	23	3
Manufacturing	13,737	92,113	1031	112	31	3	8559	59,235	290	49	10	2
Manufacture of textiles	3939	26,015	346	133	15	6	10,926	75,157	469	62	22	3
Electricity, gas, steam, and air cond. supply	6340	42,308	592	140	19	4	3124	21,570	119	55	2	1
Food and tobacco production	12,079	79,933	1106	138	51	6	6857	47,290	184	39	9	2
Non-metal mineral industry	1066	6991	135	193	6	9	337	2313	16	69	0	0
Glass–ceramic	1216	8012	113	141	5	6	432	2969	15	51	1	3
Metal processing	605	4025	39	97	4	10	117	815	1	12	0	0
Manufacture of electrical equipment	27,682	188,158	1617	86	74	4	10,389	71,692	433	60	18	3
Footwear and wood	7736	51,069	805	158	19	4	2928	20,186	88	44	4	2
Construction	74,620	492,442	7315	149	319	6	14,422	100,057	375	37	20	2
Sales and Transport												
Wholesale and retail trade	98,129	661,253	5786	88	248	4	105,004	726,646	2345	32	102	1
Hotels, camping, bars, and restaurants	38,336	255,508	2419	95	116	5	28,989	197,423	1011	51	28	1
Transportation and storage	64,579	572,259	6531	114	274	5	19,673	189,016	725	38	34	2
Credit, insurance, and services												
Insurance activities	84,831	572,727	4654	81	226	4	90,992	631,367	1905	30	78	1
Healthcare	9008	61,074	524	86	23	4	29,671	203,915	1123	55	52	3
Services	71,231	472,448	7009	148	297	6	112,747	772,230	6261	81	261	3
Other services												
Washing and dry-cleaning of textiles	1367	9186	85	93	3	3	2161	14,963	70	47	1	1
Waste collection, treatment, and disposal	2070	14,090	84	60	7	5	998	6926	21	30	0	0
Hairdressing salons	1768	11,933	91	76	8	7	7747	53,932	114	21	7	1
Cleaning services	14,888	100,304	833	83	38	4	33,270	227,446	1671	73	66	3
Gas stations	2360	15,841	137	86	7	4	426	2923	10	34	0	0

**Table 2 ijerph-20-05767-t002:** Association between sector of employment and non-accidental and accidental mortality. Men and women, Rome 2011–2019.

	Non-Accidental Mortality							Accidental Mortality						
Men	D	Age-Adjusted HR (95%CI)		Fully Adjusted HR (95%CI)				D	Age-Adjusted HR (95%CI)			Fully Adjusted HR (95%CI)		
Agriculture, forestry, and fishing	4147	0.95	0.92	0.98	0.94	0.91	0.97	158	0.99	0.84	1.17	1.00	0.84	1.19
Industry														
Steel industry	3290	0.97	0.93	1.00	0.96	0.93	1.00	185	1.06	0.91	1.24	1.08	0.93	1.27
Printing and publishing	1611	1.04	0.99	1.09	0.98	0.93	1.03	56	0.87	0.67	1.13	0.83	0.63	1.08
Pharmaceutical and chemical industry	1472	0.90	0.85	0.94	0.96	0.91	1.01	64	0.94	0.73	1.20	1.04	0.81	1.34
Manufacturing	1031	1.01	0.95	1.08	1.04	0.98	1.11	31	0.69	0.48	0.98	0.71	0.50	1.02
Manufacture of textiles	346	1.01	0.91	1.12	0.95	0.85	1.06	15	1.05	0.63	1.75	0.99	0.60	1.65
Electricity, gas, steam, and air cond. supply	592	0.90	0.83	0.98	0.91	0.84	0.99	19	0.67	0.43	1.05	0.71	0.45	1.12
Food and tobacco production	1106	1.16	1.09	1.23	1.11	1.05	1.18	51	1.24	0.94	1.64	1.19	0.90	1.57
Non-metal mineral industry	135	1.02	0.86	1.21	1.03	0.87	1.22	6	1.15	0.52	2.56	1.17	0.52	2.61
Glass–ceramic	113	1.06	0.88	1.28	1.00	0.83	1.20	5	1.11	0.46	2.66	1.04	0.43	2.51
Metal processing	39	1.66	1.21	2.27	1.48	1.08	2.02	4	3.24	1.21	8.64	2.91	1.09	7.77
Manufacture of electrical equipment	1617	0.89	0.85	0.94	0.92	0.88	0.97	74	0.92	0.73	1.17	0.99	0.78	1.25
Footwear and wood	805	1.19	1.11	1.28	1.10	1.02	1.18	19	0.67	0.43	1.06	0.61	0.39	0.96
Construction	7315	1.15	1.12	1.18	1.11	1.08	1.14	319	1.14	1.01	1.28	1.08	0.95	1.22
Sales and Transport														
Wholesale and retail trade	5786	1.06	1.03	1.09	1.02	0.99	1.05	248	0.92	0.80	1.05	0.88	0.76	1.00
Hotels, camping, bars, restaurants	2419	1.16	1.11	1.21	1.09	1.05	1.14	116	1.11	0.92	1.34	1.02	0.84	1.23
Transportation and storage	6531	0.99	0.97	1.02	0.96	0.94	0.99	274	0.99	0.87	1.13	0.99	0.87	1.13
Credit, insurance, and services														
Insurance activities	4654	0.91	0.88	0.93	0.98	0.95	1.01	226	0.93	0.81	1.07	1.02	0.89	1.18
Healthcare	524	1.00	0.92	1.09	1.09	1.00	1.19	23	0.92	0.61	1.39	0.98	0.65	1.48
Services	7009	0.92	0.90	0.95	0.98	0.95	1.01	297	0.94	0.83	1.07	0.96	0.84	1.09
Other services														
Washing and dry-cleaning of textiles	85	1.05	0.85	1.30	1.03	0.83	1.28	3	0.79	0.26	2.46	0.77	0.25	2.39
Waste collection, treatment, and disposal	84	1.09	0.88	1.35	1.01	0.81	1.25	7	1.52	0.72	3.20	1.33	0.63	2.80
Hairdressing salons	91	0.91	0.74	1.11	0.79	0.64	0.97	8	1.64	0.82	3.29	1.36	0.68	2.73
Cleaning services	833	1.42	1.33	1.52	1.25	1.17	1.34	38	1.15	0.83	1.59	0.98	0.71	1.35
Gas stations	137	1.10	0.93	1.30	1.03	0.87	1.22	7	1.12	0.53	2.35	1.01	0.48	2.12
	**Non-Accidental Mortality**							**Accidental Mortality**						
**Women**	**D**	**Age-Adjusted HR (95%CI)**			**Fully Adjusted HR (95%CI)**			**D**	**Age-Adjusted HR (95%CI)**			**Fully Adjusted HR (95%CI)**		
Agriculture, forestry, and fishing	3584	0.92	0.89	0.96	0.96	0.92	1.01	147	0.87	0.72	1.04	0.88	0.72	1.09
Industry														
Steel industry	462	0.92	0.84	1.01	0.92	0.83	1.01	23	1.08	0.71	1.65	1.11	0.72	1.69
Printing and publishing	478	1.03	0.94	1.13	0.99	0.90	1.09	21	1.10	0.71	1.70	1.08	0.70	1.67
Pharmaceutical and chemical industry	454	0.94	0.86	1.03	0.95	0.86	1.04	23	1.16	0.76	1.75	1.14	0.75	1.73
Manufacturing	290	1.03	0.92	1.16	1.02	0.90	1.14	10	0.88	0.47	1.64	0.82	0.44	1.55
Manufacture of textiles	469	0.95	0.86	1.04	0.87	0.80	0.96	22	1.13	0.74	1.73	1.04	0.68	1.61
Electricity, gas, steam, and air cond. supply	119	0.95	0.79	1.14	0.95	0.79	1.14	2	0.37	0.09	1.47	0.39	0.10	1.57
Food and tobacco production	184	1.06	0.92	1.23	1.01	0.87	1.17	9	1.22	0.63	2.36	1.20	0.62	2.33
Non-metal mineral industry	16	0.91	0.56	1.49	0.92	0.56	1.51	0	-			-		
Glass–ceramic	15	0.91	0.55	1.52	0.88	0.53	1.46	1	1.56	0.22	11.11	1.42	0.20	10.13
Metal processing	1	0.39	0.06	2.78	0.38	0.05	2.69	0	-			-		
Manufacture of electrical equipment	433	1.04	0.94	1.14	1.01	0.92	1.11	18	1.14	0.72	1.83	1.12	0.70	1.79
Footwear and wood	88	1.11	0.90	1.37	1.06	0.86	1.31	4	1.36	0.51	3.65	1.35	0.50	3.63
Construction	375	0.97	0.88	1.08	0.96	0.86	1.06	20	1.28	0.82	2.00	1.27	0.81	1.98
Sales and Transport														
Wholesale and retail trade	2345	0.97	0.93	1.02	0.92	0.88	0.96	102	1.03	0.84	1.28	1.00	0.80	1.24
Hotels, camping, bars, restaurants	1011	1.17	1.10	1.25	1.12	1.05	1.19	28	0.71	0.49	1.04	0.64	0.44	0.95
Transportation and storage	725	1.04	0.96	1.12	1.04	0.96	1.12	34	1.21	0.85	1.70	1.19	0.84	1.68
Credit, insurance, and services														
Insurance activities	1905	1.01	0.96	1.06	1.00	0.95	1.05	78	0.93	0.74	1.19	0.89	0.70	1.14
Healthcare	1123	1.05	0.99	1.11	1.03	0.97	1.10	52	1.15	0.87	1.52	1.13	0.85	1.50
Services	6261	0.97	0.94	1.00	1.00	0.97	1.03	261	0.94	0.81	1.09	0.98	0.84	1.14
Other services														
Washing and dry-cleaning of textiles	70	0.94	0.74	1.19	0.88	0.69	1.11	1	0.32	0.05	2.30	0.30	0.04	2.16
Waste collection, treatment, and disposal	21	1.50	0.98	2.30	1.41	0.92	2.16	0	-			-		
Hairdressing salons	114	1.01	0.84	1.22	0.89	0.74	1.07	7	1.42	0.67	3.01	1.29	0.60	2.78
Cleaning services	1671	1.23	1.17	1.30	1.17	1.11	1.23	66	1.14	0.88	1.46	1.11	0.86	1.44
Gas stations	10	1.62	0.87	3.02	1.59	0.85	2.96	0	-			-		

Fully adjusted models: adjusted for age, marital status, education, and place of birth.

## Data Availability

For privacy reasons, restrictions apply to the availability of the data. Individual data are accessible following strict rules on Latium region servers and cannot be exported. Aggregated data are available from the authors upon request.

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
