# Peer review of "Sector of Employment and Mortality: A Cohort Based on Different Administrative Archives"

_ijerph, 2023, doi:10.3390/ijerph20105767_

Round 1
Reviewer 1 Report
Dear authors,
Your manuscript is interesting but I need you to answer some questions:
INTRODUCTION
- The introduction is very short. The concepts necessary to understand the manuscript are not explained. For example, the authors do not discuss employment regulations on accidents at work or prevention.
- Authors should include some epidemiological data to support their claims. For example, they first talk about mortality and accidents at work at the global level. Then they go on to talk about mortality rates in Italy. The authors should compare equivalent data.
MATERIALS AND METHODS
Setting, Study Design, and Data Sources
- The authors must specify the research design.
- The authors should explain how they linked the longitudinal study data to the INPS database. If the data are anonymised I don't understand how they can link them.
Outcome and other variables
- The authors at the beginning of the manuscript discussed occupational accidents and accidental mortality. I do not understand why they are now looking at non-accidental mortality. It would be interesting to calculate both mortality rates and compare them.
Ethical considerations:
- Have you consulted the ethics committee? The authors must mention and say the reference.
DISCUSSION
- The authors have not explained how to apply the research to clinical practice.
Limitations
- The authors say they cannot establish the cause-effect of length of employment. Longitudinal studies do allow causality to be established. The authors should review the statistical analyses.
REFERENCES
- Many bibliographies are obsolete. The bibliographic citations used are more than 5 years old (43.5 %; websites do not count). The authors must update and arrange the bibliography.
- Some references are incomplete or have errors. The authors should review this section.
Author Response
We thank the reviewer for his/her comments and fruitful suggestions. Please find below a point-by-point response to each comment and question.
INTRODUCTION
Reviewer 1’s comment 1: The introduction is very short. The concepts necessary to understand the manuscript are not explained. For example, the authors do not discuss employment regulations on accidents at work or prevention.
Answer: Following the reviewer’s suggestion, we deepened the introduction, including more information on the Italian regulation on accidents at work and their prevention; their impact in terms of occupational accidents and diseases; a more detailed motivation of the study including the lack of occupational information in mortality and hospitalization data able to disentangle the contribution of work-related outcomes from those of the general population, particularly for new and emerging work-related occupational risks. As a consequence, we believe that the introduction now contains useful information for the readers.
Reviewer 1’s comment 2: Authors should include some epidemiological data to support their claims. For example, they first talk about mortality and accidents at work at the global level. Then they go on to talk about mortality rates in Italy. The authors should compare equivalent data.
Answer: In this new version of the introduction, we have added some occupational epidemiological data for Italy and three new references. We have written the first paragraph on global data, while we have focused the second paragraph on Italian data.
MATERIALS AND METHODS
Setting, Study Design, and Data Sources
Reviewer 1’s comment 3: The authors must specify the research design.
Answer: Following the reviewer’s request, we have now specified the design at line 110.
Reviewer 1’s comment 4: The authors should explain how they linked the longitudinal study data to the INPS database. If the data are anonymised I don't understand how they can link them.
Answer: The Latium Region Longitudinal Study and the INPS data were linked at the individual level with social security records based on Tax ID codes, in accordance with the Italian regulations on privacy. Neither researchers in charge of the Latium Longitudinal Study nor researchers who analyzed the data presented here had access to tax id codes linked with health data. The linkage was performed by the person, within the Department of Epidemiology, in charge of data anonymization. We have better explained the matter at lines 131-136.
Outcome and other variables
Reviewer 1’s comment 5: The authors at the beginning of the manuscript discussed occupational accidents and accidental mortality. I do not understand why they are now looking at non-accidental mortality. It would be interesting to calculate both mortality rates and compare them.
Answer: Following such comment, we have added accidental mortality as outcome under study, revising the presentation of results.
Ethical considerations:
Reviewer 1’s comment 6: Have you consulted the ethics committee? The authors must mention and say the reference.
Answer: The Latium Longitudinal Study is part of the National Statistics Program (PSN 2017–2019, LAZ-00006), which is authorized by the Italian Data Protection Authority. Since the Italian Data Protection Authority is the central authority on personal data protection, the approval of Italian Data Protection Authority acts as an ethics committee approval. Hence, there was no need for an ethical committee opinion. We have better specified it in the Institutional Review Board Statement
DISCUSSION
Reviewer 1’s comment 7: The authors have not explained how to apply the research to clinical practice.
Answer: The aim of the study was to investigate the association between the occupational sectors and accidental and non-accidental mortality. According to our point of view, results have the potential to confirm work-related outcomes and to identify emerging ones. These results mainly add value to epidemiological surveillance. Major findings will help to monitor and control spread of diseases among workers as well as the development of appropriate preventive plans. Further, the information deriving from our study may enhance the need for a thorough occupational history within the clinical management of such diseases. We added a sentence in the Discussion section.
Limitations
Reviewer 1’s comment 8: The authors say they cannot establish the cause-effect of length of employment. Longitudinal studies do allow causality to be established. The authors should review the statistical analyses.
Answer: Unfortunately, we had the information about the sector of employment from 1974 to 2011 (the baseline of our cohort). In order to estimate dose-response associations, we should have had the information of employment during the follow-up, otherwise reverse causation might occur. We have better explained this matter in the Methods section. Following the reviewer’s suggestion, we will try to obtain the occupational data from 2011 to 2019 in future projects.
REFERENCES
Reviewer 1’s comment 9: Many bibliographies are obsolete. The bibliographic citations used are more than 5 years old (43.5 %; websites do not count). The authors must update and arrange the bibliography.
Answer: Following the reviewer’s suggestion, we have rearranged the references.
Reviewer 1’s comment 10: Some references are incomplete or have errors. The authors should review this section.
Answer: Following the reviewer’s comment, we have corrected the references.
Reviewer 2 Report
The current manuscript presents an analysis of the associations between occupational/industrial sites and mortality by linkage of an Italian administrative cohort with previous occupational registry data. Apart from the associations, the manuscript focuses on the usefulness of the data linkage.
This is an interesting paper and I can basically follow the conclusion. Nevertheless, I would recommend to consider the following major and minor aspects:
Major:
1. The introduction, in particular the first paragraph, should more clearly introduce to the problem. The link between the initially referenced job-related accidents and illnesses (not part of the analysis) and insufficient mortality statistics is not straightforward. The rationale to investigate non-accidental causes of death should be explained. (and, with regard to methods, how was non-accidental defined?)
2. Throughout the manuscript – also including abstract, tables and figures, and supplemental material – it is insufficiently indicated, to which type of occupational categorisation (or reference category) the results refer, i.e. the ‘ever/never’-distinction or the ‘lifetime prevalence’.
If I understood correctly, the main results were based on the ever/never distinction of employment in an industrial sector. If so, then the description of the study population does not refer to the individual characteristics of the study participants, but individuals may be included in several occupational sectors. This difference, obvious when comparing tables S1 and S3, should be made clear and possible implications should be discussed.
3. There is no mentioning of missings in the occupational data. What was the proportion of missing occupational years in general and – if possible – for each occupational sector (only applicable for the lifetime prevalence approach)? As the inclusion criterium is just >= 1 year employment in a private occupational sector, were results based on only few years of occupational exposure? Implications of this should be discussed (misclassification?). Possibly, this could suggest sensitivity analysis with stricter inclusion criteria.
4. Maybe, an analysis (SMR / HR) of the excluded workers in the public sector could be interesting.
5. The discussion on associations of specific occupations and cancer sites (lines 289ff.) should be more related to the study’s results, the discussion overall could be a little more concise.
6. Sometimes the results section includes discussion of results, but this should be rather part of the discussion section. Accordingly, the explanation of the reference category (lines 256ff.) should be moved from the results to the methods section.
Minor:
Abstract:
1. Standardisation was made by age, so this should be added in line 25
2. Sentence in line 34 needs to be revised
3. Abbreviations should be introduced (also in the tables, manuscript)
Methods:
4. Please clarify if ‘Rome longitudinal study’ is the same as ‘Latium Region longitudinal study’
Results:
5. The summary of table S1 (lines 157ff.) seems a bit too lengthy.
6. Results for figure 2 refer to mean/SD, but I would presume the box plots display median/IQR? The text should rather refer to the content of the figure.
7. In line 28, maybe ‘statistically’ is missing?
8. Reference to table S6 is missing
9. Figures 2 and 3: label of y-axis is missing and whiskers/bars should not be truncated
10. The asterisk for the corresponding footnote is missing in tables 3, S6
11. In table S2, the columns are not labelled. Anyway, the table seems redundant, because data are already included in figure 2.
12. Table S3: The ‘N’s of the single occupational categories do not sum up to the total number?!
13. Possibly, in table S4, rather than displaying both, mean/SD as well as median and other percentiles, the above-mentioned proportions of missing occupational years (since start of job history) could be included, e.g. instead of mean/SD.
Discussion:
14. The results of the NOCCA study could be a suitable reference. (Pukkala 2009 Acta Oncol, https://doi.org/10.1080/02841860902913546), and there may be more studies.
Author Response
REVIEWER 2
Reviewer comment : The current manuscript presents an analysis of the associations between occupational/industrial sites and mortality by linkage of an Italian administrative cohort with previous occupational registry data. Apart from the associations, the manuscript focuses on the usefulness of the data linkage.
We thank the reviewer for his/her comments and fruitful suggestions. Please, find below a point-by-point response to each comment and question.
This is an interesting paper and I can basically follow the conclusion. Nevertheless, I would recommend to consider the following major and minor aspects:
Major:
Reviewer 2’s comment 1: The introduction, in particular the first paragraph, should more clearly introduce to the problem. The link between the initially referenced job-related accidents and illnesses (not part of the analysis) and insufficient mortality statistics is not straightforward. The rationale to investigate non-accidental causes of death should be explained. (and, with regard to methods, how was non-accidental defined?)
Answer: We have thoroughly modified the introduction, also by addressing some comments of reviewer 1. We have included framework information about the impact of occupational accidents and diseases in Italy; a more detailed motivation of the study including the lack of information in mortality and hospitalization data able to disentangle the contribution of work-related outcomes from those of the general population, particularly for new and emerging work-related occupational risks; new literatures references. We have added accidental mortality as outcome, and, in the Methods section, we have cited the ICD-9 codes used for accidental and non-accidental mortality.
Reviewer 2’s comment 2: Throughout the manuscript – also including abstract, tables and figures, and supplemental material – it is insufficiently indicated, to which type of occupational categorisation (or reference category) the results refer, i.e. the ‘ever/never’-distinction or the ‘lifetime prevalence’.
If I understood correctly, the main results were based on the ever/never distinction of employment in an industrial sector. If so, then the description of the study population does not refer to the individual characteristics of the study participants, but individuals may be included in several occupational sectors. This difference, obvious when comparing tables S1 and S3, should be made clear and possible implications should be discussed.
Answer: We thank the reviewer for having interpreted the draft correctly. We have tried to facilitate the reading by introducing titles in the Results section:
3.1 Characteristics of the study population
3.2 Results from statistical analyses using ever/never worked in a sector as exposure
3.3 Results from statistical analyses using the lifetime prevalent sector as exposure
3.4 Results from sensitivity analyses
Reviewer 2’s comment 3: There is no mentioning of missings in the occupational data. What was the proportion of missing occupational years in general and – if possible – for each occupational sector (only applicable for the lifetime prevalence approach)? As the inclusion criterium is just >= 1 year employment in a private occupational sector, were results based on only few years of occupational exposure? Implications of this should be discussed (misclassification?). Possibly, this could suggest sensitivity analysis with stricter inclusion criteria.
Answer: By definition, there were no missing values on occupational data because we selected only those with at least 1year of employment in a private occupational sector.
We have added a sentence in the Discussion section on the possible misclassification pointed out by the reviewer (lines 422-424). Moreover, we have conducted a sensitivity analysis stratifying the population in those who worked at least 10 years in a sector vs. those with <10 years of work (lines 190-195)
Regarding the reviewer’s request on the proportion of missing occupational years, unfortunately we cannot answer appropriately. We collected the occupational data without the single dates of start and ending of the work, but with the overall duration only. In future projects, we will try to collect the information with each temporal segment of work.
Reviewer 2’s comment 4: Maybe, an analysis (SMR / HR) of the excluded workers in the public sector could be interesting.
Answer: We have tried to follow the reviewer’s suggestion. We have calculated the age adjusted mortality rates in the population of the Rome Longitudinal study, not included in this study. The excluded population is composed by public sector workers, but also by housekeepers, professionals, unemployed, and unable to work. Since the excluded population is composed by workers and not-workers, the mortality rates in the excluded subjects are higher than those of included population, and do not allow a thorough understanding of the matter. To overcome this problem, we have selected only those who were working at the baseline (i.e., at the 2011 census) and calculated on this subset the age-adjusted mortality rates in included and excluded subjects. The results, not included in the manuscript, showed similar age-adjusted mortality rates for included and excluded populations.
Reviewer 2’s comment 5: The discussion on associations of specific occupations and cancer sites (lines 289ff.) should be more related to the study’s results, the discussion overall could be a little more concise.
Answer: following the reviewer’s comment, we have largely revised the discussion
Reviewer 2’s comment 6: Sometimes the results section includes discussion of results, but this should be rather part of the discussion section. Accordingly, the explanation of the reference category (lines 256ff.) should be moved from the results to the methods section.
Answer: following the reviewer’s request, we have moved the sentence to the Methods section.
Minor:
Abstract:
Reviewer 2’s comment 7: Standardisation was made by age, so this should be added in line 25
Answer: Done
Reviewer 2’s comment 8: Sentence in line 34 needs to be revised
Answer: Sorry! Done. See lines 33-35.
Reviewer 2’s comment 9: Abbreviations should be introduced (also in the tables, manuscript)
Answer: Done
Methods:
Reviewer 2’s comment 10: Please clarify if ‘Rome longitudinal study’ is the same as ‘Latium Region longitudinal study’
Answer: Done, see lines 114-118
Results:
Reviewer 2’s comment 11: The summary of table S1 (lines 157ff.) seems a bit too lengthy.
Answer: We have shortened the description of table S1
Reviewer 2’s comment 1: Results for figure 2 refer to mean/SD, but I would presume the box plots display median/IQR? The text should rather refer to the content of the figure.
Answer: We thank the reviewer for spotting the mistake. Indeed, we calculated both m/sd and median/IQR, but we were not consistent in reporting the results. We have now specified that the figure shows median and IQR; in addition, we have shortened the description.
Reviewer 2’s comment 13: In line 28, maybe ‘statistically’ is missing?
Answer: Sorry, we do not understand in which line it is missing.
Reviewer2’s comment 14: Reference to table S6 is missing
Answer: We thank the reviewer for spotting the omission. We have now renumbered all tables and figures.
Reviewer 2’s comment 15: Figures 2 and 3: label of y-axis is missing and whiskers/bars should not be truncated
Answer: We thank the reviewer for spotting the omission. We have now added the y-axis’label. We agree that the plot should not be truncated. However, the huge scale made the graph difficult to read. We opted for truncating the bars, with a sign indicating the truncation.
Reviewer 2’s comment 16: The asterisk for the corresponding footnote is missing in tables 3, S6
Answer: We thank the reviewer for spotting the omission. We have now included the asterisk.
Reviewer 2’s comment 17: In table S2, the columns are not labelled. Anyway, the table seems redundant, because data are already included in figure 2.
Answer: We thank the reviewer for spotting the redundance. We have now eliminated the table.
Reviewer 2’s comment 18: Table S3: The ‘N’s of the single occupational categories do not sum up to the total number?!
Answer: The reviewer is right. Indeed, we had a sector and its two different subgroups (“hairdressers and cleaning” which included both hairdressers and cleaning operators). To avoid misinterpretations, we have maintained the sector of hairdressers separated by the sector of cleaning, and we have eliminated the sector of hairdresser and cleaning together.
Reviewer 2’s comment 19: Possibly, in table S4, rather than displaying both, mean/SD as well as median and other percentiles, the above-mentioned proportions of missing occupational years (since start of job history) could be included, e.g. instead of mean/SD.
Answer: Unfortunately, we do not have the missing occupational years.
Discussion:
Reviewer 2’s comment 20: The results of the NOCCA study could be a suitable reference. (Pukkala 2009 Acta Oncol, https://doi.org/10.1080/02841860902913546), and there may be more studies.
Answer: Done

Round 2
Reviewer 2 Report
The revision has definitely improved the manuscript and all issues have been addressed (as far as possible).
Optionally, the tables for sensitivity results could be added to the supplement.